# Extraction Optimization and Structural Characteristics of Chitosan from Cuttlefish (*S. pharaonis* sp.) Bone

**DOI:** 10.3390/ma15227969

**Published:** 2022-11-11

**Authors:** Sulfath Hakkim Hazeena, Chih-Yao Hou, Jing-Huei Zeng, Bo-Heng Li, Tzu-Chih Lin, Cai-Sian Liu, Chi-I Chang, Shu-Ling Hsieh, Ming-Kuei Shih

**Affiliations:** 1Department of Seafood Science, College of Hydrosphere, National Kaohsiung University of Science and Technology, Kaohsiung 81157, Taiwan; 2Hong Yu Foods Company, Limited, Kaohsiung 806042, Taiwan; 3Department of Biological Science and Technology, National Pingtung University of Science and Technology, Pingtung 91201, Taiwan; 4Graduate Institute of Food Culture and Innovation, National Kaohsiung University of Hospitality and Tourism, Kaohsiung 812301, Taiwan

**Keywords:** cuttlefish, chitin/chitosan, reaction surface method (RSM), ^1^H NMR, FTIR

## Abstract

In fish processing, reducing the waste rate and increasing the economic value of products is an important issue for global environmental protection and resource sustainability. It has been discovered that cuttlefish bones can be an excellent resource for producing attractive amounts of chitin and chitosan. Therefore, this study optimized chitosan extraction conditions using response surface methodology (RSM) to establish application conditions suitable for industrial production and reducing environmental impact. In addition, Fourier-transform infrared spectroscopy (FTIR), ^1^H NMR and scanning electron microscope (SEM) characteristics of extracted chitosan were evaluated. The optimum extraction conditions for chitosan from cuttlebone chitin were 12.5M NaOH, 6 h and 80 °C, and the highest average yield was 56.47%. FTIR spectroscopy, ^1^H NMR, and SEM identification proved that the chitosan prepared from cuttlefish bone has a unique molecular structure, and the degree of deacetylation of chitosan was about 81.3%. In addition, it was also confirmed that chitosan has significant anti-oxidation and oil-absorbing abilities. This research has successfully transformed the by-products of cuttlefish processing into value-added products. The process not only achieved the recycling and utilization of by-products but also enhanced industrial competitiveness and resource sustainability.

## 1. Introduction

The marine fishery is the primary source of economy for many countries around the world. More than 60% of fish have considered by-products during processing. Different by-products, including fish heads, skin, fins, internal organs, fish bones, and fish eggs, cause serious disposal problems [1]. These by-products contain a large amount of protein, gelatin, etc. [2], and most are processed into low-value animal feed, fish meal or fertilizers [3]. Currently, many researchers have developed methods to extract by-products of fish processing into functional foods, nutraceuticals, and bioactive compounds [4]. For example, Taiwan Tilapia (*Oreochromis niloticus*) fishbone fermentation has reported developing anti-fatigue dietary supplements [5]. About 2.5% of all seafood is made up of cephalopods. Global cephalopods caught have risen relatively to 1961 levels by 416%, reaching a record high of about 4 million tonnes in 2013 [6]. According to the Fisheries Department of the Agriculture Committee of the Taiwan Executive Yuan, the production value of Taiwan’s cephalopod fishery and the value of cephalopod edible aquatic products in 2020 were about 59,497 metric tons/USD 158,098,000 and 82,951 metric tons/USD 22,230,887,000, respectively [7]. By-products of S. pharaonis processing, such as skin, are rich in protein, and cuttlebone is rich in calcium and other minerals. Despite the apparent economic importance of this species, little is known about the reuse of *S. pharaonis* processing by-products in the past. Cuttlefish skin gelatin, chitosan and polysaccharides are some of the value-added products processed by *S. pharaonis* [8,9]. The cuttlebone is a highly porous hard tissue that functions as a rigid buoyant tank in the animal. Its skeleton is an inorganic–organic complex composed of aragonite, protein, and β-chitin [10]. The processing of *S. pharaonis* results in the production of other value-added chemicals such as cuttlefish skin gelatin, chitosan, and polysaccharides [8,11].

Chitin is a linear polysaccharide composed of β(1 → 4)-linked 2-acetoamido-2-deoxy-β-D-glucose, which is the second most abundant polysaccharide after cellulose [12,13]. Chitin has three crystal forms: α-, β- and γ-form. Chitin is mainly used to produce chitosan through deacetylation reactions in an alkaline medium. Chitin is usually isolated from the exoskeletons of crustaceans, shrimp, and crabs in α-form [14]. Squid and cuttlefish are other important sources of chitin, where it exists in the beta form, which is more susceptible to deacetylation. The hydrogen bond between β-chitin is weaker than that of α-chitin, so it has higher solubility, reactivity, and affinity [15]. Studies have shown the potential of nanosized β-chitin as a wound-healing nanomaterial in animal studies [16].

β-Chitin can be extracted from the cuttlefish [17] and deacetylated using alkaline hydrolysis to produce more applicable chitosan [18]. Shushizadeh et al. (2015) reported the preparation of chitosan from cuttlefish [19]. Applications for chitosan include cosmetics, agriculture, food, biomedicine, wound dressings, drug delivery, and textiles, as chelating agents and refinement of industrial effluents [20]. Because of their biocompatibility, derivatives of β-chitin are often used as materials for oral and dental treatments [21]. The partially or entirely removed acetyl group from chitin to form an amine group is called chitosan. Since the amine group (-NH_2_) on chitosan is protonated in an acidic solution, chitosan becomes a positively charged glycan electrolyte, which is beneficial to solubility and biological activity [22]. Due to the bioactive properties of β-chitin and chitosan, studies on the extraction and characterization of β-chitin from squid and cuttlefish bones have become more frequent in recent years [23,24]. Two methods are commonly used to obtain chitin: chemical and biological (microbial). Compared with many environmentally friendly chemical extraction methods [25], traditionally strong acid/alkali methods for extracting chitin have shortcomings and bring various environmental problems. The fermentation process for the production of chitin from crustacean waste uses biological methods because of the easy-to-handle, simple and fast advantages. They are also usually controllable by optimizing process parameters, ambient temperature, and negligible solvent consumption, thereby reducing environmental impact and cost. However, it has so far only been limited to laboratory-scale studies [26,27].

Ramasamy et al. (2014) reported that chitosan from *Sepia kobiensis* (Hoyle 1885) bone was extracted with 40% NOH, the extraction rate was about 43.77%, and a series of structural identification and antioxidant activity analysis was performed [24]. Most of the chitosan needs to be extracted with multiple complicated steps for a long time, and the chemical extraction process is often used at high temperature and with strong alkali to improve the deacetylation effect. The higher the degree of deacetylation of chitosan, the higher the solubility in acidic solutions and the better the applicability. Therefore, the most appropriate NaOH concentration, treatment time, and reaction temperatures have to be optimized to extract chitosan efficiently. The optimal reaction conditions help to reduce environmental pollution by minimizing the use of high temperatures and high concentration of alkali with maximum chitosan yield.

Increased disposal costs of by-products and subsequent environmental pollution remain a problem during cuttlefish processing. Therefore, it is essential to use the cuttlefish by-products to generate high-value-added products. Optimizing the extraction conditions is essential for the efficient recovery of chitosan. For this purpose, the relationship between different extraction conditions has to be well-studied. The response surface method (RSM) model has effectively identified correlations between many independent and response variables while reducing the number of runs and resource utilization. The relationship between NaOH concentration, incubation time, and temperature on chitosan yield has been analyzed using the RSM model here. The optimization of extraction conditions will help to valorize the by-products effectively.

## 2. Materials and Methods

### 2.1. Reagents and Materials

Cuttlefish (*Sepia pharaonis*) bones were collected from a local cuttlefish processer (Hong Yu Foods Co., Ltd., Kaohsiung, Taiwan). The average size of cuttlefish caught in the winter season is 53–59 cm in length and 20–31 cm in width, the wet weight when received is ~5 kg, and the dry weight of the cuttlefish bone is ~80 g. The cuttlefish bones were buried in crushed ice during transportation and stored at −80°C until use. 2,2’-azino-bis(3-ethylbenzothiazoline-6-sulfonic acid) (ABTS), potassium bromide, ascorbic acid, potassium peroxydisulfate, trichloroacetic acid, and hydrogen chloride were purchased from Sigma-Aldrich (St. Louis, MO, USA). Deuterium oxide and acetic acid-d4 were obtained from ACROS (Morris Plains, NJ, USA). 1,1-diphenyl-2-picrylhydrazyl (DPPH), sodium hydrogen phosphate, sodium dihydrogen phosphate, Ferrous chloride and hydrogen peroxide were obtained from Merck Co. (Darmstadt, Germany). Other chemicals used were of analytical grade.

### 2.2. Preparation of Cuttlebones (Sepia pharaonis *sp.*)

Cuttlebones (*S. pharaonis* sp.) were collected and washed with water to remove excess salt and surface stain. The bones were broken into pieces and dried at 60 °C for 48 h. The dried bones were pulverized with a pulverizer and sieved with a 30-mesh sieve. The water activity of the sample was measured and found to be less than 0.36.

### 2.3. Extraction of Chitin and Conversion into Chitosan

For chitin extraction from cuttlebones, demineralization, deproteinization, and decolorization were carried out as described by Takiguchi (1991) [28]. Demineralization was performed by adding 100 g of cuttlebone powder to 1000 mL of 10% (*w*/*w*) HCl (1.04 M) for 24 h at 25 °C. After filter paper filtration, the residue was washed with distilled water until neutral. The residue was then immersed in 1000 mL of 10% (*w*/*w*) NaOH (2.5M) for 24 h at 60 °C for deproteinization. Proteins were removed using filtration, and the residue was washed with distilled water until neutral. The solution became colorless and transparent. Then, the cuttlebones were processed twice using the above procedure. Ethanol-soluble substances were removed and dehydrated using 250 mL of 95% ethanol to obtain crude chitin. Chitin (200 g) was dried overnight at 50 °C using an air oven. According to Sahu, Goswami, and Bora (2009), chitin obtained from cuttlebone powder was converted to chitosan through a deacetylation process using 40% NaOH aqueous solution and microwave technology [29].

### 2.4. Preparation of Chitosan and Response Surface Methodology Experimental Design

A fixed amount of chitin was immersed in a 30× volume of NaOH solution (solid-to-liquid ratio of 1:30, *w*/*v*) and put in a high temperature (50~100 °C) water bath for a standing reaction (3–9 h) to prepare chitosan. According to the three-factor experimental design, different concentrations of NaOH (7.5, 12.5, 20M) and different times (3, 6, 9 h) and temperatures (50, 80, 100 °C) were selected, respectively (Table 1). According to Gokula et al. (2018), a 3-level-3-factor (Box–Behnken) design, with three replicates designed at the center point and with 15 experiments, was performed to evaluate optimal extraction conditions for chitosan extraction. There were three samples per experiment. The effect of the combination parameters in optimizing chitosan extraction using RSM has been studied previously [30]. RSM is a statistical tool used to build an empirical model and to find the best combination of variables for the desired response. It includes three-level factorial design, central composite design (CCD), Box–Behnken design (BBD), and D-optimal design. BBD experimental design has excellent predictability among all these designs. In this work, BBD was employed by varying three variables at three levels (−1, 0, +1). The variable (0) was considered as the central point, (−1) as low level (below central point), and (+1) as high level (above central point). Independent variables and their different choice of levels are shown in Table 1. The range of these variables was obtained from the results of the initial experiments. To evaluate the effect of independent variables on chitosan extraction, 15 experiments were performed in triplicate, and the interaction of independent variables A (NaOH, M), B (time, h), and C (temperature, °C) were analyzed. The efficiency of the model and the statistical significance were analyzed using F-test and the R-test. The effect of the individual variable has been analyzed by performing a detailed analysis of variance (ANOVA) on the coded level of variables. The RSM and experiment design were performed using Design-Expert (Stat-Ease, Inc., Minneapolis, MN, USA).

### 2.5. Fourier Transform Infrared Spectroscopy (FTIR) Analysis

Chitin and chitosan were analyzed using the transmission technique. FTIR analysis was performed by grounding 2 mg sample with 100 mg of dried potassium bromide (KBr) and compressed to a 3 mm diameter disk. The IR spectra of the disc were read by a Perkin-Elmer RXI infrared with a Perkin-Elmer RXI infrared spectrophotometer. The absorbance was measured between 400 and 4000 cm^−1^ [31].

### 2.6. Preparation, Identification and Degree of Deacetylation (DD) of Chitosan by Proton Nuclear Magnetic Resonance (^1^H NMR)

The kinetics of chitosan in solution was determined by ^1^H NMR relaxation [longitudinal (T_1_) and transverse (T_2_) relaxation times and NOE] as the degree of acetylation (DA), and degree of polymerization, temperature, concentration and ionic strength. This analysis indicates that chitosan is a semi-rigid polymer with higher flexibility at higher DA, consistent with reduced electrostatic repulsion between protonated amino groups [32]. The ^1^H NMR method was used to determine the structural characterization and degree of deacetylation (DD) of chitosan. A brief description of the experimental steps is as follows: after dissolving the chitosan in the solvent (HOD) containing 1% acetic acid-d_4_ (10 mg/mL) at 70 °C, 1 mL of the chitosan solution was transferred into a 5 mm NMR tube. ^1^H NMR spectra were recorded on a Varian-Unity-Plus-400 spectrometer operating at 400 MHz. Chemical shifts (δ) are reported in ppm and referenced to the residual solvent signal of D_2_O at δ_H_ 4.80.

The integrals of the characteristic signals were used to calculate the degree of deacetylation (DD). The DD value is calculated according to the following formula:$$\mathrm{DD}\%=\left\{1-\left[I_{\mathrm{CH}3}/\left(3\times I_{H1-\mathrm{GlnNAc}}\right)\right]\right\}\times 100$$
where the I_H1-GlnNAc_ is the integral for H1 (GlcNAc) and I_CH3_ is the integral for—CH3 signal.

### 2.7. Scanning Electron Microscopy

The surface morphology of chitosan was carried out using a field scanning electron microscope (FSEM, JEOL, JSM-6330TF, Peabody, MA, USA). For SEM, the samples were gold-sputtered and observed at an accelerating voltage of 5 kV.

### 2.8. DPPH Free Radical Scavenging Ability Test

The free radical scavenging activity was determined using the 2,2-Diphenyl-1-picrylhydrazyl (DPPH) method. The sample’s hydrogen atom or electron donation ability was measured using a methanol solution of 2,2-diphenyl-1-picrylhydrazyl (DPPH). According to Wang et al. (2019) [33], 20 μL of the sample solution was taken, 180 μL DPPH reagent was added, the mixture was kept in the dark for 30 min, and the absorbance value was measured at 517 nm with a spectrophotometer. The absorbance of the sample was measured with 1000-ppm ascorbic acid as the positive control group. The free radical scavenging activity was expressed as the percentage inhibition of DPPH, which is calculated by the following formula:Inhibition percentage (%) = [(A_0_ − A_A_)]A_0_ × 100(1)

A_0_ and A_A_ are the absorbance values of the blank and test samples, respectively.

### 2.9. Reducing Power Assay

Reducing power assay was performed according to Wang et al. (2019) [33]. A total of 1 mL of 200 mM Phosphate buffer solution (pH 6.6), 1 mL of the sample solution, and 1% potassium ferricyanide were mixed together. The mixture was kept at 50 °C for 20 min, followed by rapid cooling. Later, 1 mL of 10% Trichloroacetic acid (TCA) was added. After centrifugation at 3000× *g* for 10 min, 100 μL of the supernatant was taken, and 100 μL of DW and 100 μL of 0.1% Ferric chloride (FeCl_3_‧6H_2_O solution prepared with 3.5% hydrochloric acid solution) was added. Finally, the solution was mixed evenly for 10 min, and absorbance at 700 nm was measured. The higher the absorbance value was indicated, the stronger the reducing power was shown.

### 2.10. Free Radical Scavenging Ability of ABTS

For ABTS assay, 7 mM ABTS (2,2′-azinobis [3-ethylbenzothiazoline-6-sulfonic acid]) solution was prepared, and 2.45 mM potassium persulfate solution was added. The solution was mixed well and stored at room temperature in a dark environment for 12–16 h. The stable blue-green ABTS^+^ cationic radical aqueous solution were diluted with PBS to obtain the absorbance value of 0.70 (±0.02) at 734 nm. Different concentrations of sample solutions were prepared using DMSO. A total of 10 μL of sample solution was mixed with 195 µL ABTS radical solution. The peroxidase, ABTS, and H_2_O_2_ were mixed evenly and placed in a dark room to react for 1 h to generate stable blue. The absorbance was measured at 734 nm with a spectrophotometer, and calculated by the formula as follows:Clearance (%) = [1 − (A734 nm Sample/A734 nm Blank)] × 100(2)

### 2.11. Absorption Capabilities for Oils

Oil absorption tests were performed according to Li et al. (2018) with modifications [34]. The weighted samples were soaked in soybean oils for 5 min to ensure the absorption equilibrium. The absorption capacity was measured and calculated by the formula as follows:Oil absorption (%) = [m(0) − m(R)/ weighted samples] × 100(3)
m(0) (g) and m(R) (g) are the weight of soybean oil before and after (residure), respectively.

## 3. Results and Discussion

### 3.1. Fourier-Transform Infrared Spectroscopy (FTIR) Analysis of Chitin from Cuttlefish Bones

For Fourier-transform infrared spectroscopy (FTIR) identification of chitin samples, 150 mg of KBr and 6 mg of samples were dried for 24 h and then ground into fine powder with an agate mortar. Samples were made as discs with a pressure of 5 tons and measured at wavenumbers of 400–4000. Figure 1 shows the N-acetylation sensitive variation in the considered region of the chitin spectrum, and it is also found that the bridging oxygen stretch band at 1121 cm^−1^ in the IR spectrum is the most suitable reference band. The CH-deformation band at 1470 cm proved to be the detection band. The CH-transformation mentioned here is related to the methyl group of the N-acetyl group and is therefore well suited for determining N-acetylation. The spectrum (Figure 1) shows that the absorption band at 1470 cm^−1^ is similar to the previous report [35]. The absorbance ratios of the amide II-band at 1552 cm^−1^ was determined analogous to the method of Sannan, Kurita, Ogura, and Iwakura (1978) for transmission IR spectra [35]. Beil et al. (2012) reported that the bridge oxygen stretching band at 1163 cm^−1^ in the infrared spectrum is the most suitable reference band and the CH-deformation band at 1383 cm^−1^ is a sound detection band [36].

### 3.2. RSM-Optimized Extraction of Chitosan

The chitin samples extracted above were taken for response surface method (RSM) optimization experiments. A three-level factorial design (BBD) was performed with NaOH (A), Time (B), and Temp (C) to determine the effect of chitosan yield (Y). Taking these into account, 12.5 M NaOH, 6 h’ time, and 80 °C gave the highest chitosan yield of 57.96% (Table 2). In the coefficients of the respective disguised phases calculated by regression, if the coefficients show a positive value, chitosan extraction conditions will be adequate. In the complex regression analysis table (Appendix A), the following equation was obtained.
Y = −1.54A + 1.74B + 1.89C + 5.30AB + 3.74AC − 5.13BC − 5.94A^2^ − 5.20B^2^ − 3.70C^2^ + 56.32(4)

According to this quadratic regression equation, the relationship between the correlation factor of the above-mentioned operating parameters and the chitosan yield is described, and the results are shown in the response surface graph and contour graph in Figure 2.

Borsagli and Borsagli (2019) demonstrated the effect of pH and ligands on the competitive complexation of cationic or anionic ions, especially for cationic complexation in alkaline media [37]. Chitosan has a 100% protonated (NH^3+^) amine group at a pH of 3.0 [38,39,40]. As pH increases, protonated amines decrease, and ionic interactions also decrease. Free electrons from amines and hydroxyl groups play an important role in the complexation of the analyte ions, with chelation predominating, thus reducing competition between oppositely charged ions. As pH increases and the medium changes from acidic to basic, the carboxylate group (-COO^−^) in the chain becomes dominant, and the amine is no longer protonated. Due to the free electrons of the amine (:NH_2_), the ionic interaction starts to increase again and turns towards the attraction of positive charges, and the amine (:NH_2_) plays an important chelating role. Here, in this study, under the combined conditions of reaction time and temperature through RSM, the optimal extraction rate of chitosan can be obtained without increasing the alkali concentration. Singh et al. (2019) [41] optimized the processing conditions for ultrasonic-assisted extraction of squid chitosan using a central composite design (CCD) of the reaction surface method (RSM). Squid material was sonicated at 1:18 (solids: solvent ratio) for 41.46 min at 69% amplitude and yielded 34% (*w*/*w*) chitin with a small amount of residual protein. Compared to traditional methods with 5 h extraction time, ultrasonic treatment was shown to be effective in reducing the extraction time of chitin from squid [41].

Another study investigated the chitosan from shrimp and crab shells through the chemical process of desalination, deproteinization, and deacetylation. BenSeghir and Benhamza (2017) pointed out that the coefficient of the ANOVA regression equation obtained through the response surface method (RSM) model study was as high as R^2^ = 0.9853, which showed the accuracy of the RSM model design process. The highest value of DD% of tin-derived chitosan was more than 98% after treatment under the optimized NaOH concentration of 28.6%, temperature of 81.15 °C, and reaction time of 9.55 h [42]. Deacetylation of chitin using alkaline hydrolysis yields more applicative chitosan components because amino groups in the structure are more soluble in aqueous acidic media [18]. At the same time, Shushizadeh et al. (2015) reported that the preparation of chitosan from cuttlefish [19] was similar to this study’s results. Therefore, this study’s results align with the application of industrial production needs. In the future, when extraction conditions are applied to an industrial scale, 7.5M NaOH is actually one of the suitable choices based on the consideration of environmental protection and economic benefits.

### 3.3. Chitosan Identification

The FTIR spectrum of the chitosan can be seen in the absorption band at 1411 cm^−1^. There are obvious absorption bands in the range of 600~1200 cm^−1^. The undulating characteristics confirmed that the chitosan was successfully extracted from cuttlefish bone (Figure 3). Broad peaks at 3500 and 1650 cm^−1^ indicate less pronounced hydrogen interactions or the presence of free hydroxyl groups [43]. Varma and Vasudevan (2020) [44] reported that the FTIR spectra of Horse Mussel chitosan contained bands at 564 cm^−1^ (out-of-plane curved NH, out-of-plane curved C–O), 711 cm^−1^ (out-of-plane curved NH), 1174 cm^−1^ (C–O–C stretch), 2865 cm^−1^ (CH stretch), and 3594 cm^−1^ (-OH stretch). Vibrational modes of amide C=O stretching were observed at 1604, 1598, and 1592 cm^−1,^ and the FTIR spectra formed characteristic bands in the frequency range of 4000 to 400 cm [44].

Chitosan is a natural polysaccharide obtained by partial deacetylation of chitin and is used in many industries such as food processing, cosmetics, waste treatment, water purification, wound healing, tissue repair, drug and gene delivery [20]. Most of the physical and chemical properties of such biopolymers depend primarily on the degree of deacetylation. The deacetylation technique should be fast, accurate, and preferably independent of any known standard or a calibration curve obtained with another technique. Among many methods, the ^1^H nuclear magnetic resonance (^1^H NMR) method does not require accurate knowledge of the amount of chitosan, nor does it need to determine the purity of the sample, as long as the impurity peaks do not overlap with the peaks associated with chitosan [45]. Sample preparation requires only a few milligrams of chitosan and does not require any calibration curves or reference samples with known degrees of deacetylation.

In this experiment, as per Lavertu et al. (2003), we used the ^1^H NMR method to determine the structural characterization of chitosan [45]. The analysis results are shown in Figure 4. ^1^H NMR spectra were recorded on a Varian-Unity-Plus-400 spectrometer operating at 400 MHz. Chemical shifts (δ) are reported in ppm and referenced to the residual solvent signal of D_2_O at δ_H_ 4.80. Chitosan was dissolved in D_2_O (10 mg/mL) containing 1% acetic acid-*d4*. The signal of acetyl of chitosan appeared at 1.97 ppm (Figure 4). The spectrum shows ^1^H NMR (400 MHz, D_2_O, δ, ppm): 1.97 (COCH_3_), 3.13 (H-2), 3.87 (H-5), 3.87 (H-3, H-4, H-6), and 4.72 (H-1). The degree of deacetylation of chitosan was calculated to be about 81.3% by the integrated area of ^1^H NMR spectrum.

### 3.4. Scanning Electron Microscope (SEM) Analysis of Cuttlebone Chitosan

The images show the raw material and extracted chitin from cuttlefish bones (Figure 5A), the chitin and chitosan obtained by treatment with 12.5M NaOH, 6 h and 80 °C (Figure 5B,C), and the microscope image (Figure 5D), SEM surface structure (500×, 1500×) (Figure 5E,F). The morphology of chitosan was studied using scanning electron microscopy, and the morphology of chitosan was observed with two different magnifications in different regions of chitosan. The extracted chitosan was in the form of lamellar scales, and a highly porous structure could be seen in some areas. In some parts of chitosan, the fibrillar structure can be easily distinguished. Under 500× magnification, the samples extended from strips and aggregated into clumps, and there were few scattered and independent clumps; at 1500× magnification, it was apparent that the chitosan was not only cross-linked and stacked but also possessed delicate patterns of holes and meshes.

### 3.5. Antioxidative Ability Test and Oil Absorption Test of Chitosan

In order to investigate the antioxidant activity of chitosan extracts from cuttlebones, *in vitro* stable 1,1-diphenyl-2-picrylhydrazyl (DPPH) and 2,2′-Azino-bis (3-Ethylbenzothiazoline-6-sulfonic acid) (ABTS) free radical and reducing power test was carried out. The results of the antioxidant activity of chitosan in stalk sheath are shown in Figure 6. Figure 6A shows that the extracted samples have lower DPPH free radical scavenging abilities at 4 and 8 mg/mL, about 15% and 20%, respectively. At 16 mg/mL, the free radical scavenging abilities of the extracted was about 38%. Regarding the reducing power assay test, Figure 6B shows that the reducing ability of chitosan is not apparent at 4–8 mg/mL, but it shows a sharp increase at 8–16 mg/mL. The reducing power reaches saturation at 16–32 mg/mL and no longer rises. The reducing power of the sample at 16 mg/mL was about 6.4 times higher than that at 4 mg/mL. In the ABTS•+ scavenging ability test, Figure 6C shows that the free radical scavenging curve of the tested sample at low concentration showed a rapid rise, about 22% scavenging ability at 8 mg/mL. When the concentration reaches 32 mg/mL, the removal capacity is about 24%, showing that the saturation point is reached.

Chitosan also has antioxidant properties and is suitable for “antioxidant-polymer conjugates” [46]. The antioxidant activity of chitosan is related to the nitrogen atoms in it [47]. The main mechanism of radical scavenging reaction in chitosan is hydrogen atom transfer (HAT), single electron transfer (SET), or a mixture of the two [48]. The ABTS•+ clearance value showed that the chitosan sample from the cuttlebone had a clearance value of 24% at 32 mg/mL. Currently, there is no report on the scavenging ability of ABTS•+ of chitosan.

In addition, the inhibitory effect of the chitosan DPPH free radical assay was also moderate, and the change was similar to the result of the ABTS free radical scavenging assay. Yen et al. (2007) [49] reported that the antioxidant activity of chitosan extracted from shiitake stalks was treated with high temperature and high concentration of alkali. At the concentration of 10,000 ppm (10 mg/mL), the DPPH free radical scavenging ability of chitosan from shiitake stems was about 44.5–53.5%. These results suggest that the source, mainly the extraction method, may lead to chitosan preparations with good antioxidant activity. The chitosan prepared by alkali/high temperature/long time (12.5 M NaOH, 6 h and 80 °C) had moderately low scavenging ability of ABTS•+ and DPPH. Previous studies reported that the antioxidant activity of chitosan extracts varied with its molecular weight, degree of deacetylation, and the number of free amino groups [50]. Therefore, some fractions of the chitin derivatives isolated from *Ganoderma lucidum* showed free radical scavenging ability, which varied roughly between 33–62% (DPPH free radical scavenging) [51]. In contrast, *Ganoderma lucidum* samples purified by continuous ethanol precipitation showed good scavenging activity against DPPH free radicals, with IC50 values between 2 and 2.6 mg/mL [52], indicating that different treatment conditions would affect the antioxidant performance. Another study reported that highly deacetylated, low molecular weight chitosan has better antioxidant activity because of the number of exposed free amino groups [53].

Regarding the oil absorption test, Figure 6D shows that chitosan with three different extraction conditions has a significant oil absorption capacity of about 1.9–2.4 times (g/g), respectively. Vedaiyan and Thyriyalakshmi (2020) developed an innovative chitosan-derived biodegradable sponge with high adsorption capacity, excellent recyclability and inherent lipophilic properties to remove environmentally polluting crude oil [54]. Chitosan has been evaluated for its ability to adsorb and removed contaminants such as heavy metals, organic dyes, pharmaceutical residues, and oil spills [55,56]. However, higher hydrophilicity and limited surface area are the main disadvantages of chitosan-based oil adsorbents [57]. Thus, several modification processes, such as composite formation, grafting [58] and cross-linking [59] have been conducted on native chitosan biopolymer to improve its adsorption tendency towards various oil types.

Chitosan not only can adsorb oil but also has the physiological activity of regulating blood lipids. When male rats were fed a diet containing 0.5% chitosan, although the blood cholesterol content was not affected, high-density cholesterol would increase, and low-density cholesterol would decrease, increasing the ratio of HDL-LDL [60,61]. Chitosan reduced blood cholesterol concentration in a more pronounced manner with an increasing degree of deacetylation [60]. The greater the degree of deacetylation of chitosan, the greater the number of amine groups. Therefore, the number of groups forming positive and negatively charged fatty acids in an acidic environment is also more significant. The size of chitosan particles also affects cholesterol metabolism in the body. The experimental animals were detected to have lower cholesterol concentrations in the liver and blood in the animal feed containing 2% chitosan, with a particle size of 20–60 mesh [60]. In addition, heavy metal pollution of water sources has become one of the most severe environmental and health problems today. Borsagli et al. (2016) reported that the ability of chitosan to adsorb heavy metals in water was limited by pH, which provided the possibility of applying chitosan to water treatment by effectively combining carboxyl groups in the chitosan structure [62].

Therefore, in this study, in addition to confirming that the extract was chitosan in chemical structure [44,45], the functionality of the extract was also confirmed by the above series of free radical scavenging ability, reducing ability and oil absorption ability [46,49,51,52,53,54,55,56,63].

## 4. Conclusions

The chitin samples prepared for chitosan preparation by the thermal/alkali method have been confirmed using FTIR. The optimal extraction conditions for chitosan were obtained using the RSM mode, and the highest average yield (56.47%) of chitosan preparation could be obtained under the reaction conditions of 12.5M NaOH, 6 h, and 80 °C. FTIR and ^1^H NMR have confirmed the molecular structure of chitosan. The degree of deacetylation of chitosan was about 81.3%. Further, chitosan proves its good antioxidant property through the results of its antioxidant activity by the free radical scavenging ability on DPPH radicals and ABTS radicals apart from its reducing power and chelating ability on ferrous ions. The properties will help to increase the importance of chitosan derived from aquatic resources.

## Figures and Tables

**Figure 1 materials-15-07969-f001:**
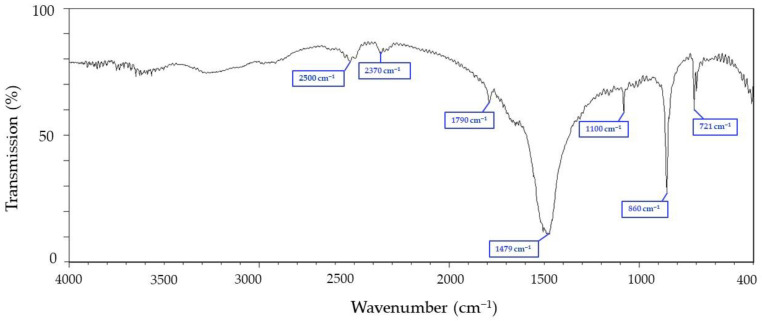
FTIR spectral analysis of chitin extracted from cuttlefish.

**Figure 2 materials-15-07969-f002:**
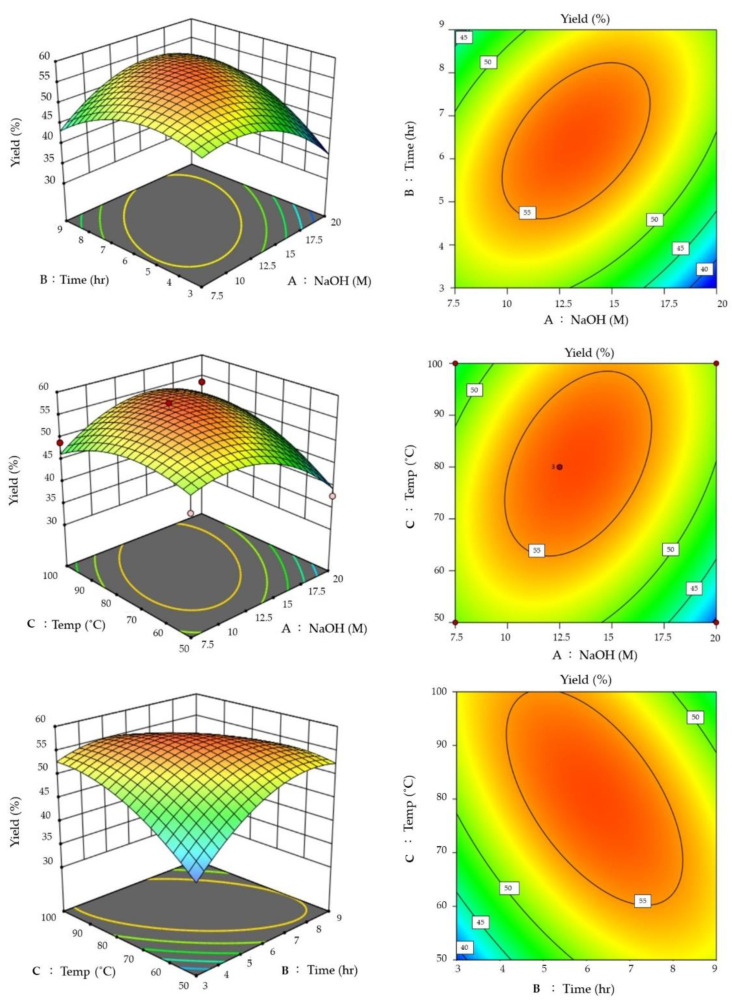
Contour plot of chitosan extraction variables. On the top of graphs are (**A**) NaOH (M) and (**B**) Time (h); on the middle of graphs are (**A**) NaOH (M) and (**C**) Temp (°C), and on the bottom of graphs are (**B**) Time (h) and (**C**) Temp (°C).

**Figure 3 materials-15-07969-f003:**
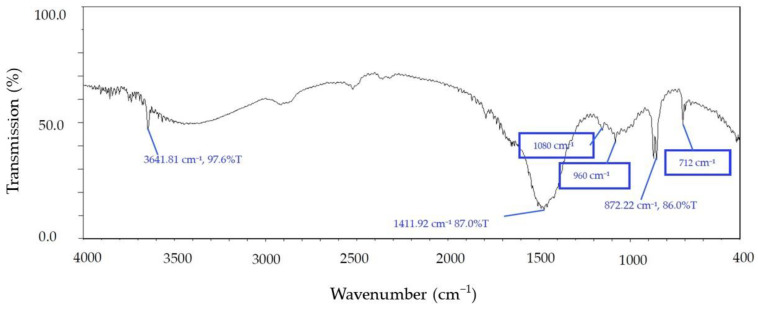
FTIR spectral analysis of chitosan extracted from cuttlefish.

**Figure 4 materials-15-07969-f004:**
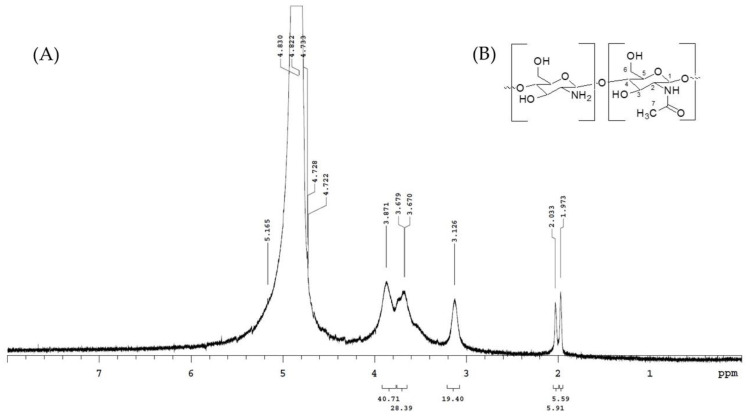
^1^H NMR identification map of chitosan from cuttlefish bone (**A**) and structure of chitosan (**B**).

**Figure 5 materials-15-07969-f005:**
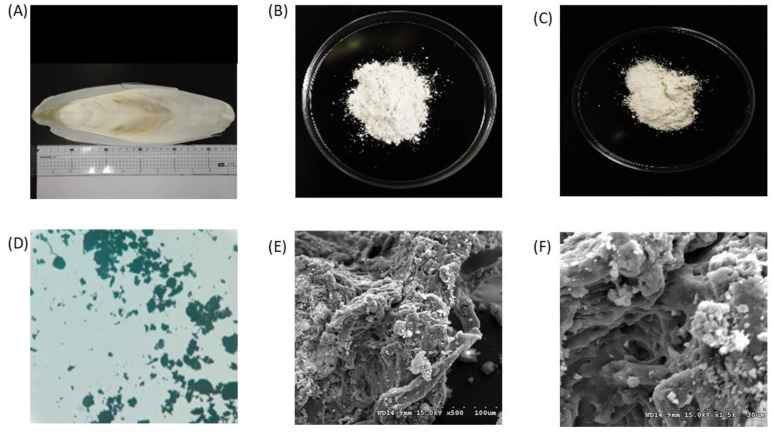
Photo of purification and preparation of chitosan by-product of cuttlefish sheath and product results. (**A**) raw cuttlefish bone material; (**B**) cuttlefish bone chitin extract; (**C**) chitosan extract from cuttlefish bone; (**D**) micrograph of chitosan from the cuttlefish bone; (**E**) SEM image (500×) of chitosan from cuttlefish bone; (**F**) SEM image (1500×) of chitosan from cuttlefish bone.

**Figure 6 materials-15-07969-f006:**
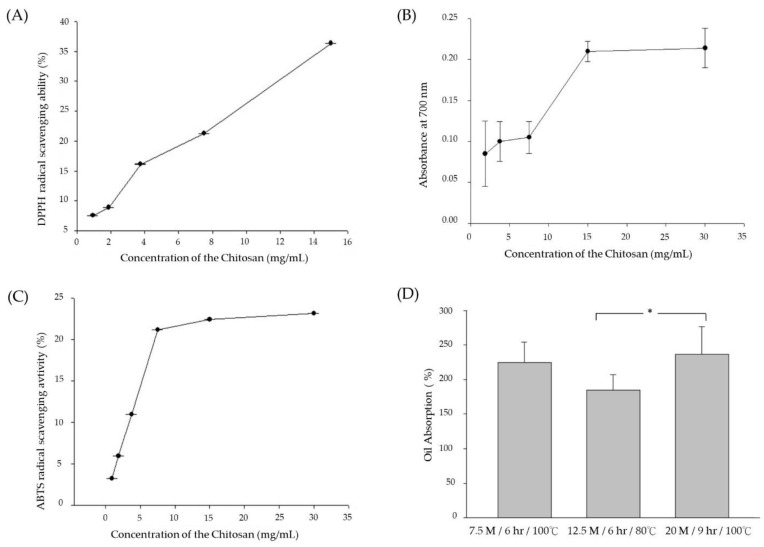
Chitosan antioxidant activity. (**A**) DPPH scavenging ability, (**B**) reducing ability, (**C**) ABTS scavenging ability, (**D**) oil absorption ability. Data were obtained from three independent experiments and are presented as mean ± SD. Data were collected and analyzed using a one-way analysis of variance and Duncan’s test. The annotation * indicates a *p*-value < 0.05.

**Table 1 materials-15-07969-t001:** Coded values and independent variables were used for optimization.

Independent Variables	Symbols	Coded Levels		
		−1	0	+1
NaOH	A	7.5M	12.5M	20M
Time	B	3 h	6 h	9 h
Temp	C	50 °C	80 °C	100 °C

**Table 2 materials-15-07969-t002:** The effect of three-level factor experimental design on the yield of chitosan.

Std	Run	NaOH (M)	Time (h)	Temp (°C)	Yield (%)
15	1	12.5	6	80	57.96
8	2	20	6	100	53.55
2	3	20	3	80	37.79
3	4	7.5	9	80	42.58
13	5	12.5	6	80	55.66
10	6	12.5	9	50	54.55
9	7	12.5	3	50	43.50
5	8	7.5	6	50	46.45
7	9	7.5	6	100	48.94
1	10	7.5	3	80	50.90
4	11	20	9	80	50.38
11	12	12.5	3	100	50.85
12	13	12.5	9	100	41.10
6	14	20	6	50	37.79
14	15	12.5	6	80	55.80

Yield (%) = (chitosan dried weight (g)/chitin dried weight (g)) × 100%.

## Data Availability

Not applicable.

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
