# Peer review of "Extraction Optimization and Structural Characteristics of Chitosan from Cuttlefish (S. pharaonis sp.) Bone"

_materials, 2022, doi:10.3390/ma15227969_

Round 1

Reviewer 1 Report (Previous Reviewer 3)

After careful revision, it could be aceeptable for publication. 

Author Response

Answers to reviewer’s comments (Round 1)

Ms. Ref. No.:  materials-2000674

Title: Extraction Optimization and Structural characteristics of Chitosan from The Cuttlefish (S. pharaonis sp.) Bone

Journal: Materials

We thank the anonymous reviewers and the editors for allowing us to improve our manuscript. The comments and suggestions were very relevant, and we have considered them all for making the revised manuscript. We feel that the quality of the manuscript has improved significantly over its previous version. All corrections have been made with tracking and changes in the manuscript, and a detailed report of the action taken has been appended below:

Reviewer#1:

Comments and suggestions

Response

1.      After careful revision, it could be aceeptable for publication.

Thanks to the reviewer's constructive suggestion to improve the quality of this manuscript, the author would give the highest respect.

Reviewer 2 Report (New Reviewer)

Ref. No.: materials-2000674

Subject: Decision on Manuscript: Extraction Optimization and Structural characteristics of Chitosan from The Cuttlefish (S. pharaonis sp.) Bone

Journal: Materials

Dear Editor,

I would like to thank you for the invite to collaborate to review process of article Extraction Optimization and Structural characteristics of Chitosan from The Cuttlefish (S. pharaonis sp.) Bone”. I recommend that is necessary a major revision of manuscript. Some comments are described below:

English should be improved in all manuscript.

Abstract: The novelty should be clearer in the abstract. Quantitative information should be added to the abstract.

Introduction: In general, the novelty of this research should be clarified. Differences among this research and another in the literature should be mentioned.

Page 2 Line 75: “. Two methods are used to obtain chitin: chemical and biological (microbial). Chemical methods are the most commonly used treatment methods commercially but are not environmentally friendly.” This phrase is not completely true, some chemical routes are environmentally friendly, then, this can not be used as a relevant disadvantage of this process.

Page 2 Line 77: “Biological extraction is a safer and cheaper method for chitin recovery, but so far, it has only been limited to laboratory-scale studies [24]. Since chitin is neither soluble in water nor weak acids or bases, its applicability might be limited.” This is nonmeaning, if biological is safer than chemical route, acid and basic medium???

Materials and methods:

All chemical reagents should be included in the manuscript, as their purity and other important parameters should be mentioned.

All units should be revised, for example, the correct unit is mL, not ml.

Page 4 Line 144: “To evaluate the effect of independent variables on chitosan extraction, 15 experiments were performed in triplicate, and the interaction of independent variables A (NaOH, M), B (Time, hr), and C (Temperature, °C) were analyzed.” How many samples for each experiment? This should be mentioned, as this is an important parameter to statistical analysis.

In the FTIR section informed that was used DRIFT method.

The methodology used to prepare the samples to NRM should be included.

Results and discussion:

Figures of FTIR and H NMR analysis should be improved.

In the FTIR analysis all term peak should be replaced to band.

Page 7 Line 238: “The reason for increasing chitosan may be that when the alkali concentration increases, 238 the solution's pH value will increase, causing OH to produce a shielding effect, protonat- 239 ing amine groups in chitosan” This not make sense. Other authors shown a different behavior for chitosan at different pH. See literatures:

10.1007/s10924-019-01449-4

Franca EF (2009) Biomolecular characterization of biopolymers in solution using computer simulation. PhD Thesis in Chemistry. Center for Science and Technology of Federal university of São Carlos, São Carlos

In the section of chitosan, the authors should be rewritten, for that they can use the literatures:

10.4028/www.scientific.net/MSF.869.750

Page 9 Line 298: “The degree of deacetylation of chitosan was calculated to be about 81.3% 298 by the integrated area of 1H NMR spectrum.” The equation used to determine the degree of deacetylation should be included in the manuscript.

The SEM methodology should be included in the section materials and methods.

More discussion about SEM analysis should be included.

Page 10 Line 315: “In order to investigate the antioxidant activity of chitosan extracts from cuttlebone, in vitro stable….” The word in vitro should be in italic font.

Figure 6 should be included error bar in all graphics.

In oil adsorption, statistical analysis should be included, for example ANOVA, chi-square. The results are very close considering the error bar, then a profound statistical should be conducted.

Conclusion should be rewritten.

References: The authors should be included more references about chitosan and chitin. In addition, they can look references in the recent years, mainly about FTIR and NMR.

Author Response

Answers to reviewer’s comments (Round 1)

Ms. Ref. No.:  materials-2000674

Title: Extraction Optimization and Structural characteristics of Chitosan from The Cuttlefish (S. pharaonis sp.) Bone

Journal: Materials

We thank the anonymous reviewers and the editors for allowing us to improve our manuscript. The comments and suggestions were very relevant, and we have considered them all for making the revised manuscript. We feel that the quality of the manuscript has improved significantly over its previous version. All corrections have been made with tracking and changes in the manuscript, and a detailed report of the action taken has been appended below:

Reviewer#2:

Comments and suggestions

Response

1.      English should be improved in all manuscript.

Response:

Thanks for your suggestions.

We have corrected the errors and resubmitted the revision with extra care.

This manuscript has been modified and undergone language editing by a native English speaker.

2.      Abstract: The novelty should be clearer in the abstract. Quantitative information should be added to the abstract.

Thanks for your suggestions.

The sentences have been deleted and rewritten.

Line 20-22; Line 27.

3.      Introduction: In general, the novelty of this research should be clarified. Differences among this research and another in the literature should be mentioned.

Thanks for your suggestions.

The sentences have been deleted and rewritten.

Line 87-89.

4.      Page 2 Line 75: “. Two methods are used to obtain chitin: chemical and biological (microbial). Chemical methods are the most commonly used treatment methods commercially but are not environmentally friendly.” This phrase is not completely true, some chemical routes are environmentally friendly, then, this can not be used as a relevant disadvantage of this process.

Thanks for your suggestions.

The sentences have been deleted and rewritten.

Line 79-85.

5.      Page 2 Line 77: “Biological extraction is a safer and cheaper method for chitin recovery, but so far, it has only been limited to laboratory-scale studies [24]. Since chitin is neither soluble in water nor weak acids or bases, its applicability might be limited.” This is nonmeaning, if biological is safer than chemical route, acid and basic medium???

Thanks for your suggestions.

The sentences have been deleted and rewritten.

Line 81-85.

Materials and methods:

6.      All chemical reagents should be included in the manuscript, as their purity and other important parameters should be mentioned.

Thanks for your suggestions.

The sentences have been rewritten.

Line 114-121.

7.      All units should be revised, for example, the correct unit is mL, not ml.

Thanks for your suggestions.

Corrected. Line 130; Line 132; Line 136.

8.      Page 4 Line 144: “To evaluate the effect of independent variables on chitosan extraction, 15 experiments were performed in triplicate, and the interaction of independent variables A (NaOH, M), B (Time, hr), and C (Temperature, °C) were analyzed.” How many samples for each experiment? This should be mentioned, as this is an important parameter to statistical analysis.

Thanks for your suggestions.

The sentences have been rewritten.

Line 149-150.

9.      In the FTIR section informed that was used DRIFT method.

Thanks for your suggestions.

The sentences have been rewritten.

Line 169.

10.  The methodology used to prepare the samples to NRM should be included.

Thanks for your suggestions.

The sentences have been rewritten.

Line 174-195.

Results and discussion:

11.  Figures of FTIR and H NMR analysis should be improved.

Thanks for your suggestions.

The figure 3 and figure 4 have been redrawn.

12.  In the FTIR analysis all term peak should be replaced to band.

Thanks for your suggestions.

Corrected.

Line 250: Line 310; Line 311; Line 315.

13.  Page 7 Line 238: “The reason for increasing chitosan may be that when the alkali concentration increases, 238 the solution's pH value will increase, causing OH to produce a shielding effect, protonat- 239 ing amine groups in chitosan” This not make sense. Other authors shown a different behavior for chitosan at different pH. See literatures: 10.1007/s10924-019-01449-4

Thanks for your suggestions.

The sentences have been rewritten.

Line 276-286.

14.  Franca EF (2009) Biomolecular characterization of biopolymers in solution using computer simulation. PhD Thesis in Chemistry. Center for Science and Technology of Federal university of São Carlos, São Carlos

Thanks for your suggestions.

The sentences have been rewritten.

Line 276-279.

15.  In the section of chitosan, the authors should be rewritten, for that they can use the literatures: 10.4028/www.scientific.net/MSF.869.750

Thanks for your suggestions.

The sentences have been rewritten.

Line 433-435.

16.  Page 9 Line 298: “The degree of deacetylation of chitosan was calculated to be about 81.3% 298 by the integrated area of 1H NMR spectrum.” The equation used to determine the degree of deacetylation should be included in the manuscript.

Thanks for your suggestions.

The sentences have been rewritten.

Line 188-195.

17.  The SEM methodology should be included in the section materials and methods.

Thanks for your suggestions.

The sentences have been rewritten.

Line 197-200.

18.  More discussion about SEM analysis should be included.

Thanks for your suggestions.

The sentences have been rewritten.

Line 349-353.

19.  Page 10 Line 315: “In order to investigate the antioxidant activity of chitosan extracts from cuttlebone, in vitro stable….” The word in vitro should be in italic font.

Thanks for your suggestions.

Corrected.

Line 365.

20.  Figure 6 should be included error bar in all graphics.

Thanks for your suggestions.

The figure 6 has been redrawn.

21.  In oil adsorption, statistical analysis should be included, for example ANOVA, chi-square. The results are very close considering the error bar, then a profound statistical should be conducted.

Thanks for your suggestions.

Corrected.

The sentences have been rewritten.

Line 388-390.

22.  Conclusion should be rewritten.

Thanks for your suggestions.

The conclusion has been modified according to the reviewer’s comments. The sentences have been deleted and rewritten.

Line 442-451.

23.  References: The authors should be included more references about chitosan and chitin. In addition, they can look references in the recent years, mainly about FTIR and NMR.

Thanks for your suggestions.

Corrected.

Line 71-73; Line 176-181; Line 313-314.

Round 2

Reviewer 2 Report (New Reviewer)

Ref. No.: materials-2000674-v2

Subject: Decision on Manuscript: Extraction Optimization and Structural characteristics of Chitosan from The Cuttlefish (S. pharaonis sp.) Bone

Journal: Materials

Dear Editor,

I would like to thank you for the invite to collaborate to review process of article Extraction Optimization and Structural characteristics of Chitosan from The Cuttlefish (S. pharaonis sp.) Bone”. My recommendation is described below:

The authors did the required all corrections and the manuscript is publishable in current version.

This manuscript is a resubmission of an earlier submission. The following is a list of the peer review reports and author responses from that submission.

Round 1

Reviewer 1 Report

This manuscript describes the processing of cuttlefish bone into chitosan and its multifunctional use. The study is promising as it targets at developing a new methodology for chitosan extraction. However, many drawbacks are still exist.

1. The materials and method section is better to be placed after the introduction section.

2. All figures are presented with very poor quality! Figures and images should be prepared in high quality format with obvious axis and embedded information.

3. The applied conditions are extremely harsh (for example, 12.5 M NaOH), is better to focus on optimizing the procedure at lower NaOH concentrations (1-5 M), especially it acquired 51 % chitosan yield at 7.5 M NaOH.  

4.The RSM-chitosan yield should be validated with experimental results such as 1HNMR. Calculations of DA should not rely only on RSM method. 

5. It is better to prove the relation between the performance of the obtained chitosan with its chemical and molecular structure (function-structure relationship).

6. References should be used while characterizing the obtained chitosan, for example within 1HNMR results (during assignment of obtained peaks), among others.

7. English editing is a MUST.

Author Response

We thank the anonymous reviewers and the editors for allowing us to improve our manuscript. The comments and suggestions were very relevant, and we have considered them all for making the revised manuscript. We feel that the quality of the manuscript has improved significantly over its previous version. All corrections have been made with tracking in the manuscript, and a detailed report of the action taken has been appended below:

Reviewer #1:

Comments and suggestions

Response

Q1. The materials and method section is better to be placed after the introduction section.

Thanks for your suggestions.

The Materials and Methods section is arranged according to the template provided by the Molecular Journal.

Q2.   All figures are presented with very poor quality! Figures and images should be prepared in high quality format with obvious axis and embedded information.

Thanks for your suggestions.

All of the Figures have been redrawn.

Q3. The applied conditions are extremely harsh (for example, 12.5 M NaOH), is better to focus on optimizing the procedure at lower NaOH concentrations (1-5 M), especially it acquired 51 % chitosan yield at 7.5 M NaOH. 

Thanks for your suggestions.

In this manuscript, we wanted to examine the optimal conditions for chitosan extraction. We used a three-factor design to extract the largest amount of chitosan as the primary study. As the reviewers valued, in the next stage of the industrial scale, the use of lower NaOH concentrations would also have appropriate economic benefits based on environmental factors.

The sentences have been deleted and rewritten.

Line 173-175.

Q4.     The RSM-chitosan yield should be validated with experimental results such as 1HNMR. Calculations of DA should not rely only on RSM method.

Thanks for your suggestions.

Chitosan yield (%) is calculated as a weight percent yield. FTIR and 1HNMR confirmed the molecular structure of the obtained product. The degree of deacetylation of chitosan was calculated to be about 81.3% by the integrated area of the 1H NMR spectrum.

The sentence has been rewritten.

Line 136; Line 213-214.

Q5. It is better to prove the relation between the performance of the obtained chitosan with its chemical and molecular structure (function-structure relationship).

Thanks for your suggestions.

The sentence has been rewritten. Line 257; Line 301-304.

Q6. References should be used while characterizing the obtained chitosan, for example within 1HNMR results (during assignment of obtained peaks), among others.

Thanks for your suggestions.

Lavertu et al. (2003) 1H NMR method have used to explain the structural analysis of chitosan. The details have provided in manuscript.

The sentence has been rewritten. Line 198; Line 211.

Q7. English editing is a MUST.

Thanks for your suggestions.

English editing has been performed by a native English speaker.

Reviewer 2 Report

The manuscript “Extraction Optimization and 1H NMR Structure Identification of Chitosan from The Cuttlefish (S. pharaonis sp.) Bone” gives some information about the production of chitosan from cuttlefish by-product. The paper is easy to read and the topic may be considered interesting, but the statistical analysis performed in the the “optimization extraction of chitosan” is not correct. This is the main drawback of the manuscript. First of all: according o the supplementary table the main parameters selected “NAOH, Time and Temperature are not significant in the extraction yield (in all the cases P>0.05), besides according to the data the applied model is not significant either! (P>0.05). Moreover, the adjusted R2 and the R2 are very different, if the model was correct these values should be more alike. This indicates that the obtained data could not be fitted by the proposed model and the surface should have been a plane.

Finally, one of the objectives of the manuscript was to “optimize” the extraction conditions of the chitosan, but no optimization was performed. In fact, only one condition was arbitrary selected. The selected condition corresponded to run 1, of Table 1, which indicated an extraction yield of 57.96%...but un run 5 this condition as repeated and the extraction obtained was significantly lower (43.38 %). This big dispersion of the data must be revised.

I have some other minor comments

Lines 278-280. What is the meaning of this paragraph?

Conclusions: The conclusions are too general and are not supported by the results.

Author Response

We thank the anonymous reviewers and the editors for allowing us to improve our manuscript. The comments and suggestions were very relevant, and we have considered them all for making the revised manuscript. We feel that the quality of the manuscript has improved significantly over its previous version. All corrections have been made with tracking in the manuscript, and a detailed report of the action taken has been appended below:

Reviewer#2:

Comments and suggestions

Response

Q1.  First of all: according o the supplementary table the main parameters selected “NAOH, Time and Temperature are not significant in the extraction yield (in all the cases P>0.05), besides according to the data the applied model is not significant either! (P>0.05). Moreover, the adjusted R2 and the R2 are very different, if the model was correct these values should be more alike. This indicates that the obtained data could not be fitted by the proposed model and the surface should have been a plane.

Finally, one of the objectives of the manuscript was to “optimize” the extraction conditions of the chitosan, but no optimization was performed. In fact, only one condition was arbitrary selected. The selected condition corresponded to run 1, of Table 1, which indicated an extraction yield of 57.96%...but un run 5 this condition as repeated and the extraction obtained was significantly lower (43.38 %). This big dispersion of the data must be revised.

Thanks to the reviewer's constructive suggestion to improve the quality of this manuscript, the author would give the highest respect.

Based on reviewer’s suggestion, we carefully looked at the raw data again and did identify where the data was miscalculated. We have updated the correct data in Table 1, recalculated the statistical calculation (Table S1) and redrawn the RSM diagram (Figure 2).

Line26; Line 133; Line 430.

Q2.    Lines 278-280. What is the meaning of this paragraph?

Thanks for your suggestions.

The paragraph has been deleted.

Q3. Conclusions: The conclusions are too general and are not supported by the results.

Thanks for your suggestions.

The conclusion has been modified according to the reviewer’s comments. The sentences have been deleted and rewritten.

Line 433-436.

Reviewer 3 Report

Authors gave us a meaningful study about Extraction Optimization and 1H NMR Structure Identification of Chitosan from The Cuttlefish (S. pharaonis sp.) Bone. It is useful for the fish industry development especially for the by-product utilization in future. However, there are some critical issues should be clarified as follows:

1.       The more detail information of cuttlefish should be provided, such as annual output in Asian or in the whole world.

2.       The previous studies about cuttlefish by-products should be cited in the part of introduction.

3.       In the part of results and discussion, the firstly appearance of FTIR should provide the whole information, although most of readers know what’s the meaning of it. Similar, there are too many abbreviation should give the whole information, such as RSM, SEM and so on.

4.       The figure 1 should be more clearly and the line should be shown more smooth.

5.       The decimal places of data in figure 1 and figure 3 should be consistent.

6.       In figure 5, the photo of (F) is not clear, please change another one.

7.       In figure 6, the reducing ability in (B) was shown as absorbance, why not consistent with (A) and (C) shown as %.

8.       In the part of 3.1, the cuttlefish information such as the size, weight, caught season, fresh or dried, these information may effect the bone as raw.

9.       The part of conclusion should concise and point out the highlights for the whole study.

Author Response

We thank the anonymous reviewers and the editors for allowing us to improve our manuscript. The comments and suggestions were very relevant, and we have considered them all for making the revised manuscript. We feel that the quality of the manuscript has improved significantly over its previous version. All corrections have been made with tracking in the manuscript, and a detailed report of the action taken has been appended below the attached file.

Round 2

Reviewer 2 Report

The authors did not response to all the comments.

The proposed model for the "optimization" has no significant terms, this indicates that the parameters time, temperature and NaOH do not have any effect on the extraction yield.

The selection of the variables was not performed correctly, instad all the parameters were included. Adding a variable to the model will always increase R 2, but this is not statisically correct as the parameters are not significant. A stepwise methodology should have been followed to determine the significant terms, and only the significant ones should have been used. I strongly suggest to revise publications related to the optimization process such as:

Marchetti, L., Argel, N., Andres, S. C., & Califano, A. N. (2015). Sodium-reduced lean sausages with fish oil optimized by a mixture design approach. Meat Science104, 67-77.

Myers, R. H., Montgomery, D. C., & Anderson-Cook, C. M. (2016). Response surface methodology: process and product optimization using designed experiments. John Wiley & Sons.

Besides, I asked" , one of the objectives of the manuscript was to “optimize” the extraction conditions of the chitosan, but no optimization was performed. In fact, only one condition was arbitrary selected" This was not properly responded as, in fact, no "optimization " was realy performed in the manuscript.

Reviewer 3 Report

All issues were addressed.